# Comparative Evaluation of STEAP1 Targeting Chimeric Antigen Receptors with Different Costimulatory Domains and Spacers

**DOI:** 10.3390/ijms25010586

**Published:** 2024-01-02

**Authors:** Yixin Jin, Claire Dunn, Irene Persiconi, Adam Sike, Gjertrud Skorstad, Carole Beck, Jon Amund Kyte

**Affiliations:** 1Department of Cancer Immunology, Institute for Cancer Research, Oslo University Hospital, 0379 Oslo, Norway; yixin.jin@rr-research.no (Y.J.); claire.dunn@ous-research.no (C.D.); irene.persiconi@rr-research.no (I.P.); adam.sike@ous-research.no (A.S.); c.a.beck@ous-research.no (C.B.); 2Department of Clinical Cancer Research, Oslo University Hospital, 0424 Oslo, Norway; 3Faculty of Health Sciences, Oslo Metropolitan University, 0130 Oslo, Norway

**Keywords:** chimeric antigen receptor, STEAP1, costimulatory domain, CAR spacer, 4-1BB, CD28, cell therapy, prostate cancer, xenograft mouse model, T cell phenotype

## Abstract

We have developed a chimeric antigen receptor (CAR) against the six-transmembrane epithelial antigen of prostate-1 (STEAP1), which is expressed in prostate cancer, Ewing sarcoma, and other malignancies. In the present study, we investigated the effect of substituting costimulatory domains and spacers in this STEAP1 CAR. We cloned four CAR constructs with either CD28 or 4-1BB costimulatory domains, combined with a CD8a-spacer (sp) or a mutated IgG-spacer. The CAR T-cells were evaluated in short- and long-term in vitro T-cell assays, measuring cytokine production, tumor cell killing, and CAR T-cell expansion and phenotype. A xenograft mouse model of prostate cancer was used for in vivo comparison. All four CAR constructs conferred CD4^+^ and CD8^+^ T cells with STEAP1-specific functionality. A CD8sp_41BBz construct and an IgGsp_CD28z construct were selected for a more extensive comparison. The IgGsp_CD28z CAR gave stronger cytokine responses and killing in overnight caspase assays. However, the 41BB-containing CAR mediated more killing (IncuCyte) over one week. Upon six repeated stimulations, the CD8sp_41BBz CAR T cells showed superior expansion and lower expression of exhaustion markers (PD1, LAG3, TIGIT, TIM3, and CD25). In vivo, both the CAR T variants had comparable anti-tumor activity, but persisting CAR T-cells in tumors were only detected for the 41BBz variant. In conclusion, the CD8sp_41BBz STEAP1 CAR T cells had superior expansion and survival in vitro and in vivo, compared to the IgGsp_CD28z counterpart, and a less exhausted phenotype upon repeated antigen exposure. Such persistence may be important for clinical efficacy.

## 1. Introduction

CAR T cell therapy is approved as standard treatment for leukemia, lymphoma, and myeloma [1,2,3]. In non-hematological cancers, efficacy has only been reported in individual cases [4,5,6]. For these cancers, immunosuppression and a lack of good target antigens are key hurdles [4,7,8]. However, promising data have recently been reported from CAR T therapy against neuroblastoma [9], and there has been some evidence of its effect on prostate cancer [10]. Also, some CAR T studies in solid cancers have indicated an effect of giving tumor escape through antigen (Ag) loss [5,6]. Interestingly, therapy with tumor-infiltrating lymphocytes can be highly effective against melanoma, suggesting a potential for cell therapy against solid cancers [11].

CAR T cell therapy is based on isolating the patient’s T cells from the blood, transducing them with a gene encoding the CAR, and giving them back to the patient. This treatment is resource-demanding and likely to be applied only for the treatment of refractory cancers. It is, therefore, crucial to target a CAR-antigen that is retained in aggressive cancer and unlikely to be lost during metastasis. In most CARs, the antigen-binding part is a single-chain fragment variable (scFv) derived from an antibody (Ab) that is then fused to the CD3zeta (CD3ζ) signaling domain from the T cell receptor (TCR). The addition of a costimulatory domain introduced in second-generation CARs led to the clinical efficacy of CAR T cell therapy [4,12].

We have developed a CAR where the scFv recognizes the protein six-transmembrane epithelial antigen of prostate-1 (STEAP1), a protein expressed in multiple cancer forms [13]. STEAP1 has been shown to be associated with tumor proliferation and progression [14,15,16,17,18]. Furthermore, several studies have indicated that STEAP1 promotes tumor invasion into various organs and the peritoneum [17,18,19]. STEAP1 is expressed in ~90% of prostate cancers and in sizeable subpopulations of other cancer types such as pancreatic cancer, glioblastoma, lung cancer, bladder cancer, breast cancer, ovarian cancer, leukemia, lymphoma, and head and neck cancer [14,15,20,21]. Furthermore, a key oncogenic driver in pediatric Ewing’s sarcoma is a chromosomal translocation giving the fused EWSFLI1 gene, which leads to the upregulation of STEAP1 [14,15,16,22,23,24].

In normal tissues, STEAP1 is mainly expressed in the prostate, which is not a vital organ and can be removed. Some reports have indicated low/moderate STEAP1 expression in other tissues, such as adipose tissue and the bladder [14,15,20]. However, across different studies, a consistent normal tissue expression of STEAP1 has only been shown in prostate tissue. The first trial with a STEAP1-targeting drug (an antibody–drug conjugate) indicated a favorable safety profile, and the side effects noted in this trial were not considered related to STEAP1-directed toxicity [25]. At ESMO in October 2023, interim results were reported from a phase 1 study of a bispecific antibody against STEAP1 in patients with castration-resistant metastatic prostate cancer [26]. Here, the authors concluded that the drug was tolerable, encouraging preliminary efficacy.

The costimulatory CAR domain confers T cells with more potent effector functions [27] and appears crucial for the persistence of CAR T cells in vivo and for their clinical activity [4,12,28]. The most commonly used costimulatory CAR domains are derived from CD28 or 4-1BB, whereas other alternatives like OX-40, GITR, and ICOS have also been explored [29,30,31]. Most studies have been conducted with CD19-targeting CARs against hematological cancers. However, the optimal costimulatory domain may differ between different cancer forms, target antigens, and CAR constructs. The binding properties of the CAR scFv also influence the signaling levels [32]. The CAR spacers should facilitate appropriate binding of the scFv to the particular target epitope and, in some cases, need to be tailored [33]. Moreover, some spacers may bind to host receptors and cause off-target activation [30,34]. In the present study, we investigated the effect of substituting costimulatory domains and spacers in the STEAP1 CAR.

## 2. Results

### 2.1. Cloning and Expression of Alternative STEAP1 CAR Constructs

We cloned four STEAP1-specific CAR T constructs based on the Oslo1 scFv by combining two alternative costimulatory domains (CD28, 4-1BB) with different extracellular spacers (sp), based on CD8a or IgG1 (Figure 1): Oslo1_IgGsp_CD28z (JK10), Oslo1_CD8sp_41BBz (JK11), Oslo1_IgGsp_41BBz (JK15), Oslo1_CD8sp_CD28z (JK16). The IgG1-spacer had four aa mutations to prevent binding to Fc-receptors, as reported by Hombach et al. [34]. All four constructs had identical transmembrane (TM) domains from CD8a. The suicide/sorter/marker gene RQR8 [35] was cloned into each construct in tandem with the CAR. RQR8 may be used for the elimination of transduced cells with rituximab. Here, we use it as a marker to identify the expression of the CAR construct. STEAP1 CAR expression after transduction of primary T cells was assessed by flow cytometry. All four STEAP1 CAR variants were expressed well on both CD4^+^ and CD8^+^ T cells, with a transduction efficacy in primary T cells ranging from 45% to 90% (Figure 1B).

### 2.2. T Cells Expressing Four Alternative CAR T Variants Specifically Kill STEAP1+ Target Cells

We investigated the ability of the four CAR T cell variants to kill tumor cells by analyzing the induction of active caspase-3 in STEAP1^+^ target cells (prostate cancer cell line 22Rv1). Apoptosis of the 22Rv1 target cells was measured by analyzing the intensity of FITC-DEVD-FMK bound to active caspase-3 by flow cytometry. Dead cells were excluded with Fixable Viability Dye, and T cells were excluded by staining for CD3. Representative contour plots are shown in Figure 1. The results showed that all four CAR T constructs enabled recipient T cells to specifically kill STEAP1+ target cells. The induction of active caspase-3 was 4-fold increased when 22Rv1 cells were cultured together with STEAP1 CAR T cells, compared to non-transduced control T cells (Figure 1).

### 2.3. CD28-CAR T Cells Produce More IFNγ and TNFα Than 41BB-CAR T Cells

We selected the most dissimilar STEAP1 CAR constructs, where both the spacer and costimulatory domain were different, for systematic comparative evaluation in vitro and in vivo. The expression of these constructs, Oslo1_IgGsp_CD28z (JK10) and Oslo1_CD8sp_41BBz (JK11), was investigated by repeated transductions (n = 17) of primary T cells from healthy donors. As shown in Figure 1, the CAR expression was high (>70%) for both JK10 and JK11, still significantly higher for JK10 (*p* = 0.048).

Next, we compared the cytokine response in the Oslo1_IgGsp_CD28z (JK10; CD28-CAR) and Oslo1_CD8sp_41BBz (JK11; 41BB-CAR) CAR T cells. As controls, we employed T cells expressing an irrelevant CD19 CAR and non-transduced T cells. The CD19 CAR is identical to JK11, except for the scFv, and resembles the construct used in tisagenlecleucel (fmc63scFv_CD8aTM/spacer_41BB-z), which is approved for use against leukemia [1]. The T cells were co-cultured overnight with tumor target cells using a panel of STEAP1-positive (22Rv1, C2-4B, LNCap) and STEAP1-negative (22Rv1 STEAP1 knockout) prostate cancer cells that we have previously described [13,36]. The production of IFNγ and TNFα was measured by flow cytometry.

A representative experiment is shown in Figure 2 and Figure 3. The results demonstrated that both of the CAR T cell variants, JK10 and JK11, produced IFNγ (Figure 2) and TNFα (Figure 3) upon encounter with all three STEAP1 positive target cell lines, but not with the STEAP1 negative controls. The IFNγ response was stronger in CD8^+^ CAR T cells, whereas the CD4^+^ CAR T cells produced more TNFα. Interestingly, T cells expressing the Igsp_CD28 CAR (JK10) generally produced more of both IFNγ and TNFα compared to the CD8sp_41BB CAR T cells (JK11). There was some variation in the response to the different target cells, as the proportion of CAR T cells responding to stimulation with 22Rv1 was usually higher than to LNCap or C2-4B. The STEAP1 levels were also higher in the 22Rv1 cells, which had been sorted for high STEAP1 expression [13].

### 2.4. CD28-CAR T Cells Kill More STEAP1+ Targets Than 41BB-CAR T Cells in Short Term Assays

The ability of CAR T cells to kill tumor cells was first assessed in overnight caspase assays, which measure the induction of apoptosis in target cells (Figure 4A,B). JK10 and JK11 CAR T cells, as well as CD19 CAR and NT T cell controls, were cultured with STEAP1-positive target cells (22Rv1, LNCap, C2-4B), or with STEAP1-negative (22Rv1 KO, C2-4B KO) or STEAP1-low (shRNA knockdown) targets. The assays demonstrated that both JK10 and JK11 CAR T cells killed the target cells in an STEAP1-specific manner. Of note, the CD28-CAR T cells (JK10) killed a significantly higher proportion of target cells.

### 2.5. Efficacy of 41BB-CAR T Cells Is Not Inferior to CD28-CAR T Cells in Long-Term Killing Assays

The target killing by CAR T cells over time was investigated using IncuCyte long-term time-lapse imaging (Figure 4C,D). The target cells, 22Rv1, were transduced by lentivirus to express the nucleus-located protein GFP. One day before co-culture with CAR-T cells, 22Rv1 target cells were seeded and irradiated. Cryopreserved T cells were added to the target cells at E:T ratios of 1:1 and 2.5:1. The number of target cells was then monitored over 8 days. We found that both JK10 and JK11 CAR T cells efficiently killed STEAP1-positive targets. More target cells were killed at the higher E:T ratio, as expected. Contrary to the short-term caspase assays (Figure 4A,B), the recorded killing by JK11 T cells (41BB-CAR) tended to be greater than for JK10 T cells (CD28-CAR).

### 2.6. 41BB-CAR T Cells Exhibit Superior Expansion in Long-Term Assay with Repeated Stimulations

CAR T cell efficacy in a patient is influenced by the long-term expansion and survival of CAR T cells and by how the CAR T cells react to repeated antigen exposure. It has been suggested that the 4-1BB costimulatory domain gives CAR T cells superior survival as compared to CD28, but this is debated and may depend on the particular construct and target. We investigated this question in vitro by subjecting the CAR T cells to six repeated stimulations from STEAP1^+^ tumor cells over 21 days while monitoring their expansion and phenotypic development (Figure 5A). Twice per week, half of the T cells were transferred to a new plate seeded with freshly irradiated target cells. The remaining T cells were then counted using flow cytometry counting beads. The results revealed distinct differences between the JK10 (CD28-CAR) and JK11 (41BB-CAR) T cells. Both JK10 and JK11 initially proliferated in response to the presence of STEAP1, with cell numbers peaking at 14 days (Figure 5B). However, JK11 CAR T cells proliferated significantly more than CAR T cells containing JK10, reaching much greater cell numbers and maintaining levels at four times their original cell number to the end of the assay (day 21).

### 2.7. 41BB-CAR T Cells Express Lower Levels of Exhaustion Markers after Repeated Stimulations

CAR T cells from the long-term assay described in the previous section were assessed for activation marker and checkpoint receptor expression twice per week by multi-parameter flow cytometry. The same phenotype characterization was performed on PBMCs from this blood donor as well as after the initial one-week stimulation/expansion necessary for retroviral CAR transduction (JK10 or JK11). As shown in Figure 5C,D, the initial stimulation and CAR transduction of PBMCs led to an increase in the CD4/CD8 ratio, upregulation of the activation marker CD25, and the checkpoints LAG3 and TIM3. This was the same for both JK10 and JK11 CAR T cells, but the upregulation of CD25, LAG3, and TIM3 was more pronounced for JK10.

The CAR T cells were then frozen–thawed and used in the assay described above, with stimulation from STEAP1 tumor cells twice per week. Over the 18 days monitored, the majority of JK10 cells were CD4^+^, whereas the proportion of CD8^+^ cells gradually increased over time for JK11, reaching >60% at day 18 (Figure 5C). The JK10 cells remained mostly (>90%) CD25 positive, suggesting an activated phenotype. The CD25 expression increased even for JK11 cells and reached about 80% in both CD4^+^ and CD8^+^ T cells from day 4. LAG3, PD1, TIGIT, and TIM3, which are markers of activation and exhaustion, were all expressed at considerably higher levels on JK10 cells compared to JK11. A large proportion of JK11 T cells remained negative for LAG3, TIM3, or PD1, whereas 70–90% of JK10 cells expressed each of these markers. TIGIT was expressed in a few CAR T cells before the first stimulation with STEAP1+ tumor cells. In JK11 T cells, the expression increased only moderately to about 30%, whereas it increased to 70–80% in the JK10 cell population. The expression of PD1, LAG3, and TIM3 was higher in the CD8^+^ than CD4^+^ subsets for both JK10 and JK11 CAR T cells.

We also investigated T cell differentiation/maturation by classifying the CAR T cells into naive, effector memory (EM), central memory (CM), or terminally differentiated (TEMRA) cells based on well-established markers (CD45RA, CCR7; Appendix A). The data showed that >95% of T cells acquired an effector memory or central memory phenotype after the initial activation with retroviral transduction. After the repeated stimulations with STEAP1+ tumor cells, the majority of both JK10 and JK11 CAR T cells acquired an EM phenotype. At the end of the assay (day 18), 90% of CD4^+^ JK10 cells and 70% of CD4^+^ JK11 cells were EM cells, as were ~65% of CD8^+^ JK10 and JK11 cells. The TEMRA percentage remained low (<5%).

### 2.8. Both 41BB-CAR T Cells and CD28-CAR T Cells Have Robust Therapeutic Efficacy In Vivo

The in vivo activity of the JK10 and JK11 CAR T cells was tested in a subcutaneous (s.c.) xenograft mouse model, in which we have previously shown that the JK11 CAR T cells have therapeutic activity [13]. NOD *scid* gamma (NSG) mice were engrafted s.c. on both sides of the hind legs with the human STEAP1-positive prostate cancer cell line 22Rv1. At days 10 and 17 post-engraftment, the mice were treated with JK10 CAR T cells, JK11 CAR T cells, or NT T cells (Figure 6). Tumor growth was monitored by both caliper measurements and bioluminescence imaging.

The results showed that both JK10 and JK11 CAR T cells had robust anti-tumor activity in vivo, whereas the NT T cells did not. These results were consistent between bioluminescence imaging (Figure 6) and caliper measurement (Appendix A). By both methods of measurement, a statistically significant difference (*p* < 0.05) was observed between the JK10 and NT groups and between JK11 and NT. There was no significant difference in the tumor signal (Figure 6) or size (Appendix A) between the JK10 and JK11 groups.

As an indication of toxicity, the body weight of the mice was measured once per week after the first treatment with T cells on day 10. There was no significant difference in body weight until the last day of measurement (day 38) in any of the groups (Figure 6D). The mice were also monitored for signs of distress (lethargy, shivering, changes in fur, and behavioral changes). No such signs were detected.

### 2.9. Tumor Infiltration by T Cells In Vivo Was Only Detected for 41BB-CAR T Cells

The infiltration of JK10 and JK11 CAR T cells into tumors was investigated by immunohistochemistry (IHC) assessment of tumors harvested 38 days after engraftment. Sections from three tumors in each group (NT n = 3; JK10 n = 3; JK11 n = 3) were stained for total T cells (anti-huCD3) and CAR T cells (CD34/RQR8 tag). Representative results are shown in Figure 7. CD3^+^ T cell and CAR T cell infiltration was observed in the JK11 CAR T cell-treated tumors but not in those treated with JK10 or NT T cells. These results indicated that JK11 CAR T cells, but not JK10 CAR T cells, were retained in the residual tumors at the end of the in vivo experiment.

## 3. Discussion

Studies of CAR T cells with various specificities, as well as the clinical experience with CD19 CAR T products, have indicated that costimulatory domains and spacers substantially influence clinical activity [4,12]. We have previously reported the development of a STEAP1 CAR (JK11) with a 41BB-costimulatory domain and transmembrane I and spacer (sp) domains derived from CD8a. This choice of design was based on the fact that a CD19 CAR with the same backbone has robust clinical activity and is approved for clinical use against B-cell cancers. In the present study, we investigated how alternative costimulatory domains and spacers in the CAR influenced the in vitro and in vivo activity of the STEAP1 CAR. The purpose was to compare the activity of the JK11 CAR construct to alternative second-generation CAR designs based on the same Oslo1 scFv before bringing the CAR into clinical development.

The results of the initial experiments showed that the four CAR constructs tested were expressed well in primary CD4^+^ and CD8^+^ T cells and that they all conferred recipient T cells with STEAP1-specific activity. For a more extensive comparison, we focused on the two most different CAR constructs, i.e., Oslo1_IgGsp_CD28z (JK10) and Oslo1_CD8sp_41BBz (JK11). We found that the JK10 CAR had a stronger IFNγ and TNFα response, as measured in overnight co-culture assays with tumor target cells. Similarly, the JK10 CAR T also showed a more potent killing capacity in short-term assays. These observations are in line with data from other CAR constructs with other specificities. In multiple in vitro studies, CARs with CD28 costimulatory domains outperform other constructs with alternative costimulatory domains [27,29]. However, the clinical breakthrough in CAR T cell therapy arrived with a 41BBz CAR with the same backbone (cd8aTM/sp-41BBz) as JK11 [37]. Several studies have suggested that the 41BBz construct has superior survival and long-term persistence upon repeated antigen exposure [12,28,31]. This may, however, vary depending on the construct and target in accordance with the binding patterns and resulting signaling.

To investigate how the STEAP1 CAR T variants responded to repeated antigen stimulation, we performed long-term assays with repeated stimulations of the JK10 and JK11 CAR T cells with STEAP1+ tumor cells. Interestingly, the JK11 T cells expanded far more over time than JK10. Both JK10 and JK11 CAR T cells acquired an activated phenotype with expression of the measured activation/exhaustion markers (CD25, PD1, LAG3, TIGIT, and TIM3). However, the JK11 cells expressed less of all these markers, and a large proportion of JK11 T cells remained negative for PD1, LAG3, TIGIT, and TIM3, suggesting a less exhausted phenotype. Moreover, the JK11 CAR T cells outperformed the JK10 cells in long-term live-imaging IncuCyte killing assays, which lasted for one week. Taken together, the in vitro data thus suggested that the JK11 CAR T cells have superior expansion and persistence upon long-term and repeated antigen exposure and also mediate more effective tumor cell killing over time.

Furthermore, we evaluated CAR T efficacy in vivo using the same mouse model in which we had previously evaluated the JK11 CAR. In this model, both the JK10 and JK11 CAR T cells had significant anti-tumor activity, and we did not observe any difference in their effect on tumor burden, as measured by IVIS and caliper. In spite of their similar anti-tumor activity, we did observe a difference in the tumor infiltration by JK10 and JK11 T cells at the end of the experiment. Persisting CAR T cells in the tumor were only detected for JK11. This observation supports the hypothesis that the 41BBz CAR T cells may have superior survival properties, which is in line with our observations in the long-term in vitro assays and other research groups that have reported similar observations with other CAR T products [12,31]. Such persistence may be important for durable clinical efficacy, including protection against tumor relapse.

In a previous study of CD19 CAR T cells, we found that CAR T cells with excellent anti-tumor activity in vitro did not work in vivo, using an NSG mouse model. This discrepancy was due to the binding of the IgG-domain to FcR in the NSG mice, which in turn caused activation-induced cell death of the CAR T cells, as well as important toxicity [30]. In the present study, the STEAP1 CAR with the mutated IgG-spacer showed activity in vivo, with no weight loss or other signs of toxicity. This observation supports the concept that the four mutations reported by Hombach et al. [34] in the Ig-derived spacer serve to prevent binding to the mouse FcR, significantly reducing AICD and toxicity in this animal model.

The JK10 and JK11 CAR constructs include the suicide/marker gene RQR8, which allows for the elimination of CAR T cells with rituximab [35]. Such a safety gene offers no guarantee that all CAR T cells can be eliminated and is not in general required for clinical development but may offer a level of increased safety. An arguably safer alternative would be transient mRNA-based retargeting, which we used in the above-mentioned CD19 CAR T study [30] and have also combined with RQR8 [38]. However, there is no evidence of clinical efficacy from mRNA-based CAR T cells, and even this approach does not protect against immediate toxicity. Both JK10 and JK11 CAR-T cells were generated using a gamma-retroviral vector. Similar gamma-retroviral delivery systems are currently used in clinically approved CAR-T products (axicabtagene ciloleucel, brexucabtagene autoleucel) [39,40]. There are as yet no clinical data from CAR T cell trials targeting STEAP1. The JK11 CAR was, to our knowledge, the first CAR developed against STEAP1; however, recently, one more STEAP1 CAR has been reported [41]. As mentioned, the target STEAP1 has been evaluated in two clinical studies with antibody-based drugs, which were considered tolerable with some evidence of clinical activity [25,26].

There are certain limitations in the study reported herein. This study was not designed to provide definite evidence regarding the superiority of any of the CAR constructs. Rather, this study represents a screening of common CAR T design variants to support the choice of a CAR for clinical development. The in vivo direct comparison has, therefore, not been repeated, and the findings from this experiment should be interpreted with caution. The JK11 CAR construct has, however, shown robust in vivo activity in multiple animal experiments [13]). In order to move the JK11 CAR towards clinical development, we are conducting further animal experiments with this construct. We used NSG mice, which are deficient in B and T cells and do not produce cytokines (common gamma chain mutation) necessary for homeostatic survival and functional activity of lymphoid cells. Therefore, to ensure the survival of our CAR-T cells for the duration of our experiments, we administered rhIL-2 twice a week. The use of rhIL2 should not be required in patients where endogenous immunological support mechanisms will be present but may be explored. As of today, the available CAR-T therapies are not used in combination with rhIL-2 in patients. These therapies, however, target antigens expressed in the hematological compartment, and it is now known if CAR T cells targeting other antigens might benefit from the systemic administration of cytokines.

## 4. Materials and Methods

### 4.1. CAR Retroviral Vector Construction

A scFv targeting STEAP1 was cloned into the SFG retroviral vector, with or without IgG-hinge extracellular domains combined with the 4-1BB or CD28 co-stimulate domains of the second-generation CAR. The suicide/sorter/marker gene RQR8 was also used, as previously described [13].

### 4.2. Retrovirus Production and CAR T Cell Transduction

Retrovirus production and CAR T cell transduction were performed as described [13]. In brief, Phoenix-Ampho HEK cells were co-transfected with packaging plasmids gag/pol and RD114 env (gift from Dr. Martin Pule) with the CAR vectors using X-tremeGENE 9 reagent (Roche, Oslo, Norway). The virus supernatants were harvested and stored at −80 °C before use. The CAR T cell transduction was performed in retronectin-coated 6-well plates (Takara, Gothenburg, Sweden) and centrifuged at 900× *g* for 60 min at 32 °C, and then incubated and expanded at 37 °C for 6–7 days. Cells were given fresh RPMI medium containing 10% FBS, 100 U/mL penicillin/streptomycin (complete medium), and rhIL-2 (100 IU/mL) every 2 days.

### 4.3. Cell Cultures and Flow Cytometry T Cell Assays

All prostate cancer cell lines, 22Rv1, LNcap, C4-2Bsh8955, C4-2Bsh9419 (both STEAP1 knockdown cell lines), and C4-2BshGFP (knockdown negative control), were gifts from Dr. Fahri Saatcioglu. Phoenix-Ampho HEK cells (ATCC, CRL-3213) and T cells from healthy donors were cultured and expanded as described previously [13]. T-cell phenotyping and functional analysis were determined by flow cytometric analysis and the data acquired on BD LSRFortessa or BD FACSymphony A5 flow cytometers (BD Biosciences, Franklin Lakes, NJ, USA). Flow cytometry data were analyzed using FlowJo software v10.1 (Tree Star, Ashland, OR, USA). T-cell intracellular cytokine and cytotoxicity assays were performed as described previously [13]. The antibodies and reagents used are listed in Appendix A. Cell viability was determined with Fixable Viability Dye efluor780 (Thermo Fisher Scientific, Oslo, Norway); target cell apoptosis was measured by the CaspGLOW Fluorescein Active Caspase-3 Staining Kit (Thermo Fisher Scientific, Oslo, Norway). The BD Cytofix/Cytoperm reagent (BD Biosciences Nordic, Oslo, Norway) was used for the intracellular staining.

### 4.4. CAR T Cell Proliferation and Phenotype under Longitudinal Repeated Target Exposure

The GFP-tagged 22Rv1 cells were seeded at 1 × 10^6^/well in 24-well plates and irradiated at 20 grays (Gy) the next day. CAR T cells and non-transduced (NT) T cells were added at 1 × 10^6^/well in a 2 mL complete medium effector to a target ratio (E:T) of 1:1. Twice per week, 1 mL of the suspended cells were transferred to a new 24-well plate seeded with freshly irradiated GFP tagged 22Rv1 (1 × 10^6^ cells/well) plus 1 mL fresh complete medium without rhIL2. The remaining suspended cells were phenotyped and counted by flow cytometry using 123count eBeads Counting Beads (Thermo Fisher Scientific) and antibodies for phenotypic markers as indicated. The target cells were gated out by the GFP tag.

### 4.5. Real-Time Killing Assay

The real-time killing assay was performed with the IncuCyte S3 live cell image system (Sartorius AG, Goettingen, Germany) as described previously [13]. The plate was screened and imaged every 3 h for 8 days. The target cells, 22Rv1, were transduced by lentivirus to express the nucleus-located GFP. The target cells were seeded at 1 × 10^5^/well in 96-well plates and irradiated at 20 Gy. The next day, cryopreserved T cells were thawed and added to the culture at an E:T ratio of 1:1 and 2.5:1. The plate was monitored in real-time and imaged by IncuCyte S3 every 3 h for 8 days. The data were analyzed using the IncuCyte S3 built-in software.

### 4.6. In Vivo Xenograft Model Tumor Killing Assay

Eleven-week-old NXG (NOD-Prkdc^scid^ Il2rg^tm1^/RJ) (Janvier Labs, Le Genest-Saint-Isle, France) immune-deficient mice were subcutaneously injected with 2 × 10^6^ 22Rv1 cells (firefly luciferase transduced) + 20% Matrigel (Corning Life Sciences, Wiesbaden, Germany) on both flanks. Ten days post-implantation, when the tumors were palpable, the mice were randomly grouped into three groups: 1 × 10^7^ per mouse NT-T cell (mice n = 3, tumor n = 6), JK10 CAR T cells (mice n = 6, tumor n = 12), and JK11 CAR T cells (mice n = 6, tumor n = 12) were injected into the tail vein. A second dose of T cells (1 × 10^7^ cells per mouse) was i.v. injected 1 week later, and 100 IU/gram body weight of rhIL-2 was given intraperitoneally (i.p.) twice per week. Tumor growth was monitored by bioluminescence imaging once per week (Xenogen Spectrum system; Grantham, UK) and measured by caliper twice per week as described previously [13].

### 4.7. Immunohistochemistry (IHC)

The formalin-fixed paraffin-embedded (FFPE) tumor tissue sections were IHC stained with anti-huCD3 (ab17143, Abcam, Cambridge, UK; 1:200) or anti-huCD34-epitope in marker gene RQR8 (ab8536, Abcam, Cambridge, UK; 1:500) as described previously [13] and imaged by a Zeiss Axiolab 5 microscope (Zeiss Norway, Oslo, Norway).

### 4.8. Statistical Analysis

Statistical analyses were performed using GraphPad Prism 9. Student *t*-tests were used for statistical comparison groups in the in vitro assays. Mann–Whitney U test was used for statistical comparisons of tumor size between two groups in the in vivo experiment. All *p*-values given are two-tailed values.

## 5. Conclusions

These findings suggest that both JK10 and JK11 CAR T cells have robust STEAP1-specific anti-tumor activity in vitro and in vivo without detectable off-target effects or toxicity. Furthermore, these results indicate superior expansion, survival, and persistence of the JK11 T cells, with less expression of phenotypic exhaustion markers compared to the JK10 variant. The leap from animal models to patients is huge, and for clinical development, it is reasonable to choose a construct resembling one with proven efficacy, as does JK11. Based on these considerations, we conclude that there is a basis for developing the JK11 STEAP1 CAR for clinical testing.

## Figures and Tables

**Figure 1 ijms-25-00586-f001:**
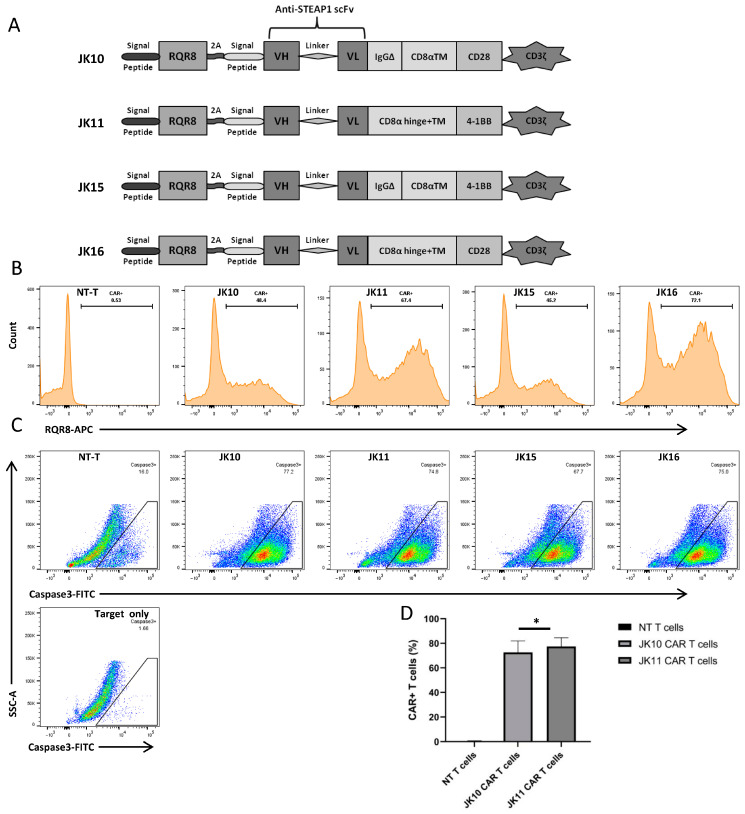
STEAP1 CAR constructs with alternative costimulatory domains and spacers. (**A**) Schematic of anti-STEAP1 CAR constructs with CD28 or 4-1BB costimulatory domains, combined with CD8a or IgG1-based spacers. All constructs share a CD8a transmembrane domain. (**B**) Representative expression of CAR constructs in primary T cells. NT-T: Non-transduced control T cells. (**C**) Killing of STEAP1+ target cells (22Rv1) by four CAR T variants and NT controls. Apoptosis of target cells was measured by analyzing the intensity of FITC-DEVD-FMK bound to active caspase-3 by flow cytometry. Representative pseudocolor plots of 22Rv1 cells after overnight co-culture with different CAR T cells at E:T = 1:1 show a gating strategy for the identification of cells positive for active caspase-3. (**D**) Expression of JK10 and JK11 in transduced T cells from 17 experiments in 8 healthy donors (* *p* = 0.048, paired *t*-test).

**Figure 2 ijms-25-00586-f002:**
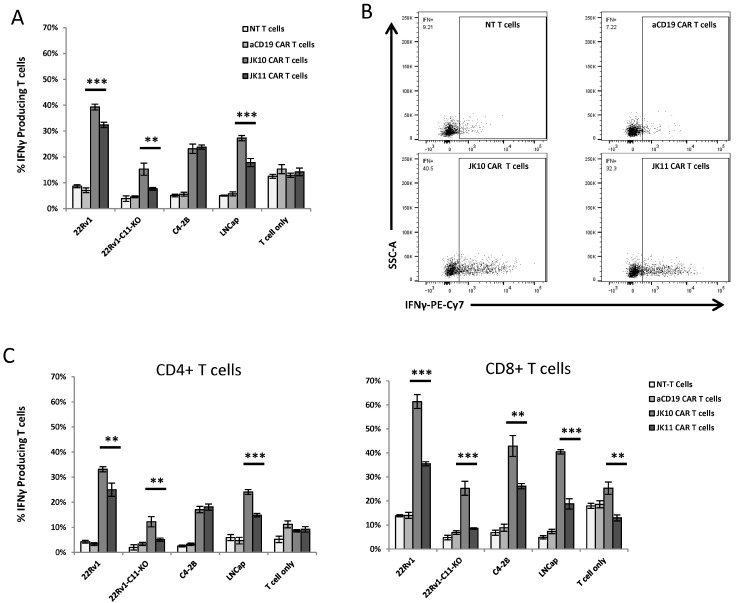
CD28 CAR T cells produce more IFNγ than 4-1BB CAR T cells upon co-culturing with STEAP1+ target cells. (**A**) Flow cytometry analysis of non-transduced (NT) T cells (white bars), CD19 CAR T cells (light gray bars), JK10 CAR T cells (dark gray bars), and JK11 CAR T cells (black bars) cultured in quadruplicates with different target cells at an effector-to-target ratio of 1:1 for 16 h. (**B**) Representative dot plots of IFNγ production by the four types of T cells co-cultured with 22Rv1 cells. (**C**) IFNγ production in CD4^+^ (**left**) or CD8^+^ (**right**) T cells. The T cells were gated on RQR8^+^ (a marker for CAR expression) or CD3, CD4, or CD8 for the non-transduced cells. 22Rv1, LNCaP, and C4-2B are STEAP1^+^ prostate cancer lines. 22Rv1-C11-KO cells are a 22Rv1 STEAP1 knockout cell line developed by CRISPR/Cas9 technology. Data in A and C are the means of quadruplicates, with error bars representing SD. ** *p* < 0.001, *** *p* < 0.0001 (*t*-test).

**Figure 3 ijms-25-00586-f003:**
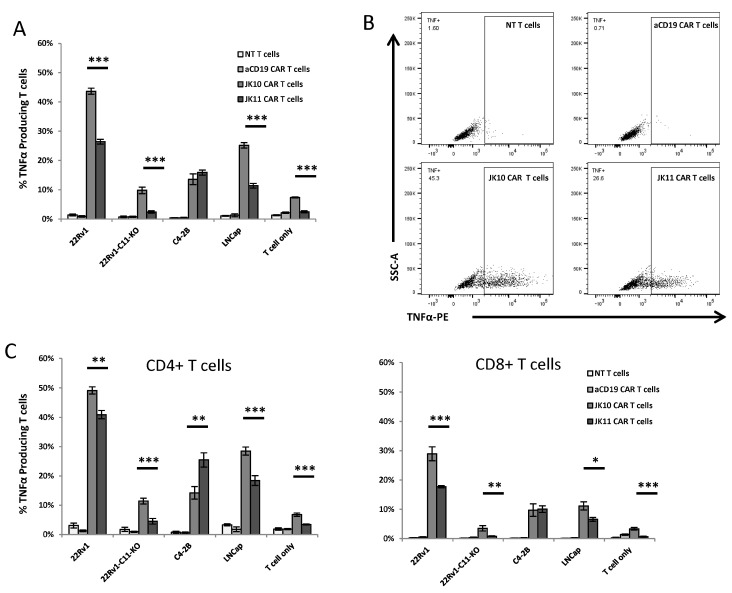
CD28 CAR T cells produce more TNFα than 4-1BB CAR T cells upon co-culturing with STEAP1+ target cells. (**A**) Flow cytometry analysis of non-transduced (NT) T cells (white bars), CD19 CAR T cells (light gray bars), JK10 CAR T cells (dark gray bars), and JK11 CAR T cells (black bars) cultured in quadruplicates with different target cells at an effector-to-target ratio of 1:1 for 16 h. (**B**) Representative dot plots of TNFα production by the four types of T cells co-cultured with 22Rv1 cells. (**C**) TNFα production in CD4^+^ (**left**) or CD8^+^ (**right**) T cells. The T cells were gated on RQR8^+^ (a marker for CAR expression) or CD3, CD4, or CD8 for the non-transduced cells. Data in A and C are the means of quadruplicates, with error bars representing SD. * *p* < 0.01, ** *p* < 0.001, *** *p* < 0.0001 (*t*-test).

**Figure 4 ijms-25-00586-f004:**
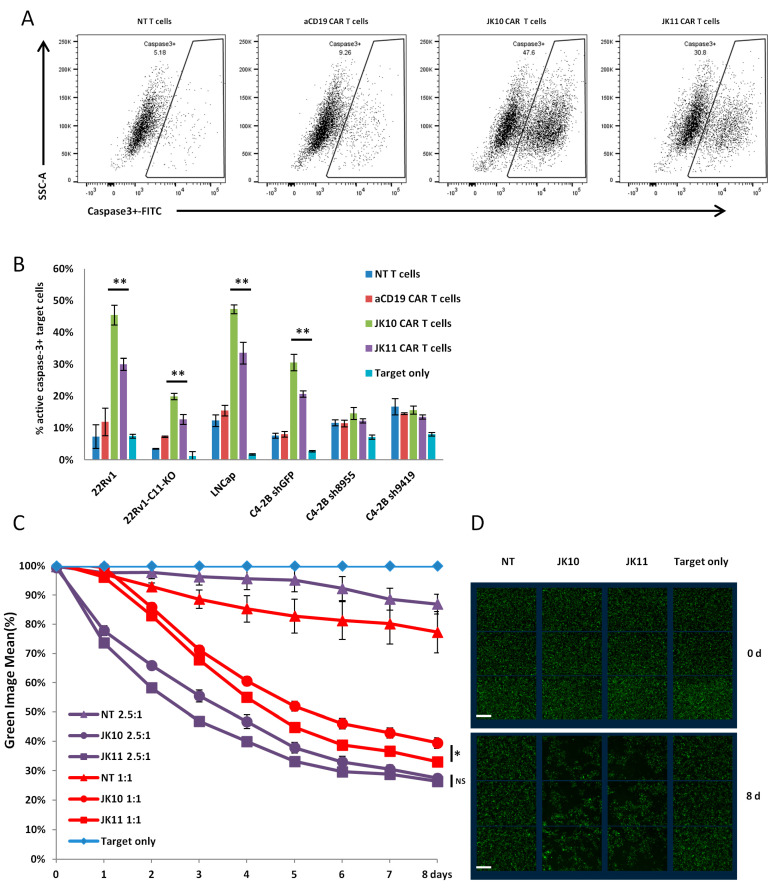
CD28 CAR T cells induce more apoptosis in STEAP1+ target cells than 4-1BB CAR T cells in short-term in vitro co-culture but do not kill more target cells in long-team co-culture assays. (**A**,**B**) Non-transduced (NT) T cells, CD19 CAR T cells, JK10 CAR T cells, and JK11 CAR T cells were cultured in quadruplicate with STEAP1-positive or knockout/down prostate cancer cell lines. The effector cells were co-cultured for 16 h with target cells at an effector-to-target (E:T) ratio of 1:1. Apoptosis of target cells was measured by analyzing the intensity of FITC-DEVD-FMK bound to active caspase-3 by flow cytometry. Target cells cultured alone were included as controls to indicate the baseline level of active caspase-3 in each cell line. (**A**) Representative contour plots of 22Rv1 cells after co-culture with effector cells, showing a gating strategy for the identification of cells positive for active caspase-3. (**B**) Percentage (mean ± SD) of apoptotic cells among STEAP1-positive and STEAP1-knockdown/out targets after co-culture with effector T cells (E:T ratio of 1:1). ** *p* < 0.001 (*t*-test). (**C**,**D**) 22Rv1 target cells were transduced by lentivirus to express the nucleus-located protein GFP. One day before co-culture with CAR-T cells, 22Rv1 target cells were seeded at 1 × 10^5^/well in 96-well plates and irradiated at 20 grays the next day. Cryopreserved T cells were thawed and added to the target cells at the indicated E:T ratios. The plate was real-time monitored over 8 days and imaged by IncuCyte S3. (**C**) Mean ± SEM GFP expression was determined by the IncuCyte S3 built-in software and normalized with the target cell-only group. (**D**) Representative images showing GPF+ 22Rv1 target cells at the start of the co-culture and on day 6. Bar scale: 300 μm; (E:T = 2.5:1), NS (not significant), * *p* < 0.01 (*t*-test).

**Figure 5 ijms-25-00586-f005:**
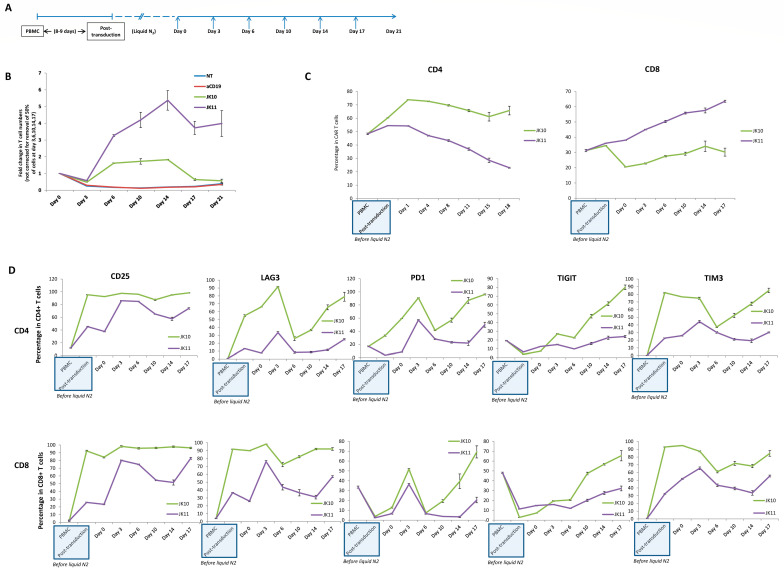
JK11 CAR T cells showed better long-term proliferation in vitro than JK10 CAR T cells. (**A**) Timeline of a long-term in vitro co-culture proliferation assay. Irradiated 22Rv1 target cells were seeded at 1 × 10^6^/well in 24-well plates. After 1 day (day 0), CAR T cells were thawed and added at 1 × 10^6^/well. Twice per week, half of the suspended cells were transferred to a new plate seeded with freshly irradiated target cells (days after the start of the co-culture were indicated by arrows). The cells were counted and phenotyped by flow cytometry twice per week. (**B**) T cell expansion measured by 123count eBeads™ Counting Beads. To separate the cell populations, the 22Rv1 cells were transduced with GFP and sorted, giving >98% GFP+ 22Rv1 cells. Error bars represent the SEM of quadruplicates. (**C**) CD4^+^ and CD8^+^ T cell fractions as a percentage of JK10 and JK11 T cells over time. (**D**) Expression of CD25, LAG3, PD1, TIGIT, and TIM3 in CD4^+^ (**top**) and CD8^+^ T cells (**bottom**). Error bars represent the SEM of quadruplicates.

**Figure 6 ijms-25-00586-f006:**
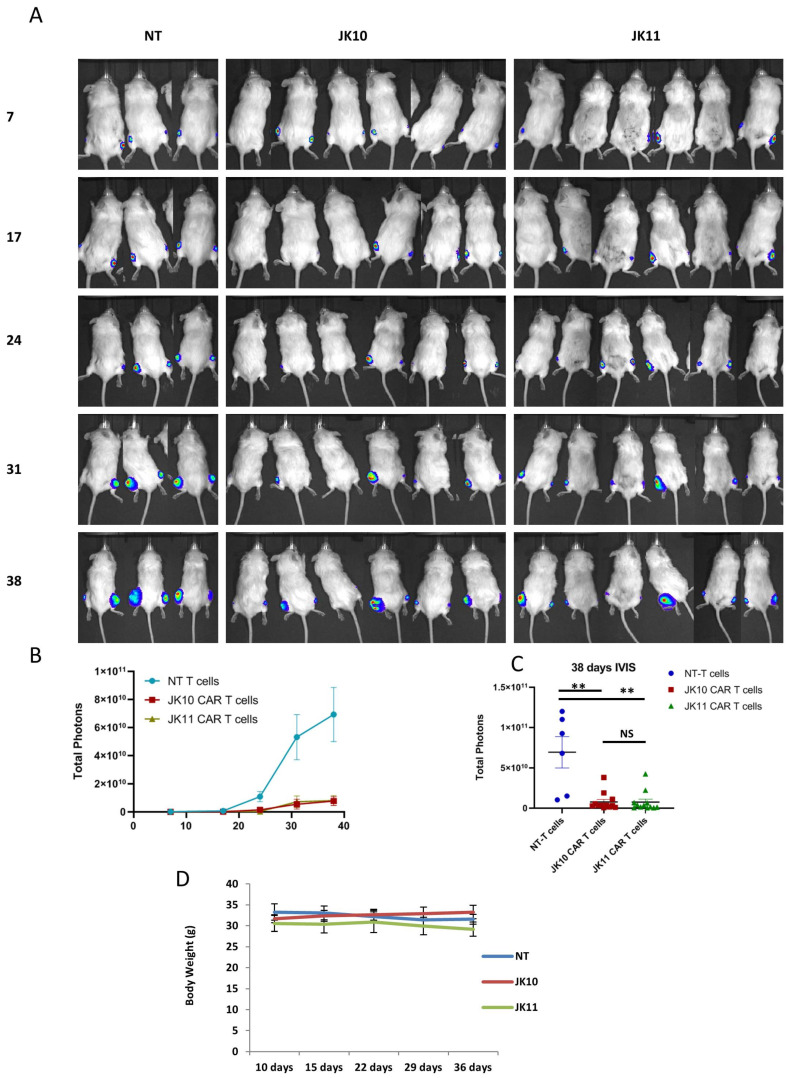
CD28 and 4-1BB CAR T cells inhibit tumor growth in vivo. (**A**) Bioluminescence images of individual mice at multiple time points in a subcutaneous tumor model. NSG mice were engrafted subcutaneously on both hind legs with 2 × 10^6^ luciferase expressing 22Rv1 prostate cancer cells. On days 10 and 17, the mice were treated with 1 × 10^7^ non-transduced (NT) T cells (N_tumor_ = 6) or CD28 CAR T cells (N_tumor_ = 12) and 4-1BB CAR T cells (N_tumor_ = 12) by intravenous injection. Bioluminescence signals of individual mice were measured at indicated time points (days after tumor engraftment). (**B**) Tumor growth is measured once a week by bioluminescence IVIS imaging. The data are presented as the mean ± SEM of tumors in each group. (**C**) Bioluminescence signals (mean ± SEM) 38 days after tumor engraftment. Statistical analyses in (**B**,**C**) were performed with the Mann–Whitney U test. NS (not significant), ** *p* < 0.001. (**D**) The body weight of the mice was measured once per week after the first treatment with T cells on day 10. Error bars indicate SD.

**Figure 7 ijms-25-00586-f007:**
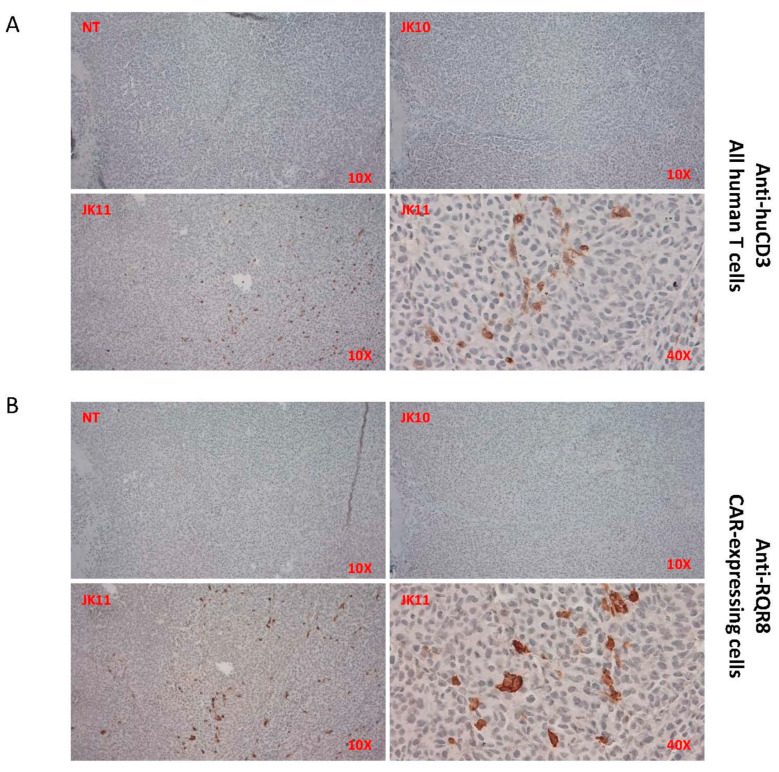
JK11 CAR T cells infiltrate tumors in vivo. (**A**,**B**) Tumors after 38-day engraftment were fixed, paraffin-embedded, and stained for IHC with anti-huCD3 (T cells) or QBen10 (CAR/RQR8-expressing cells). (**A**) aCD3 and (**B**) aRQR8, showing representative staining from each group of mice, with 10× and 40× magnification, as indicated. T cell infiltration (**A**) was only observed in the JK11 CAR T cell-treated tumors and corresponded to CAR T cell infiltration (**B**).

## Data Availability

Experimental protocols and data are available upon reasonable request.

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
