# Peer review of "Comparative Evaluation of STEAP1 Targeting Chimeric Antigen Receptors with Different Costimulatory Domains and Spacers"

_ijms, 2024, doi:10.3390/ijms25010586_

Round 1

Reviewer 1 Report

Comments and Suggestions for Authors

An interesting paper with potential implications for treating solid tumors with CAR-T cells. 

Authors previously developed a CAR against STEAP1 (13/ Mol Ther Oncolytics. 2022), which seems to be an attractive target highly specific for aggressive cancers and almost exclusively present in the prostate in normal tissues, consistently absent from other tissues. In early clinical studies, first attempts to target STEAP1 using antibody drug conjugates or a bispecific antibody against STEAP indicated a favorable safety profile and promising preliminary efficacy results. Previous work from the authors also demonstrated efficacy of the transduced CART against prostate cancer models expressing the target protein, STEAP1, in vitro and in vivo preclinical studies. In the present study, they asked whether substituting certain co-stimulatory domains and spacers in the STEAP1 CAR can improve or interfere with CAR-T persistence and efficacy in vitro and in vivo.

in vitro they found that while CD28-CAR T-STEAP1 cells produce more IFNγ and TNFα, and kill more STEAP1+ targets than 41BB-CAR T-STEAP1 cells in short term assays, 41BB-CAR T-STEAP1 cells exhibit superior expansion in long term assay with repeated stimulations, coinciding with lower levels of exhaustion markers after repeated stimulations. In vivo, in the conditions tested, both 41BB-CAR T cells and CD28-CAR T cells had robust anti-tumor efficacy in vivo, with no significant differences found in the tumor signal or the tumor size. Interestingly, in contrast to CD28-CAR T-STEAP1, 41BB-CAR T cells were found to be persistent in the tumors. These results are in line with various CART studies comparing 41BB vs CD28-based CART cells showing higher short term effects for CD28-based CART cells, and longer effects for 41BB-based CAR-T cells. These findings could help in the choice of CART cells to be used in the context of solid tumors.

Globally an extensive, carefully executed and well controlled piece of work. Different models and appropriate controls were used for in vitro and in vivo testing. However, a number of caveats exist that should be addressed prior to publication:

1- I have some concerns about the use of recombinant IL2 in the experiments, which could mask/interfere with the effects and characteristics of the different CAR-Ts (beyond the CAR-T production phase), particularly in the in vivo experiments.

A- Can the authors specify why they inject rIL2 in Mice ? Do they have data to present without such supplementation ? Is rIL2 also added in short and long term in vitro assays? If so, authors should clarify this point in the method section. 

B- Also authors should discuss the rationale for introducing IL2 into mice with regards to the perspective of clinical testing in patients

2- The in vitro part is far more complete and detailed than the in vivo part. I'm aware that conducting new experiments can be time-consuming if they don't have the data in hand, but at least the authors could discuss better the in vivo part and expand to include a bit more with regard to potential experimental biases (IL2 etc…), propose experiments to be carried out to better validate/expand the results obtained in vitro (rechallenge experiments, characterize activation/exhaustion states of injected cells, play on the doses of injected CART cells, the number of injections, the number of mice tested …), to be indeed in the limitations section and/or elsewhere.

Author Response

Please see the attached response letter

Reviewer 2 Report

Comments and Suggestions for Authors

The authors provided a thorough investigation into the optimization of chimeric antigen receptor (CAR) design for targeting the six-transmembrane epithelial antigen of prostate-1 (STEAP1). The study is meticulously conducted, employing a range of in vitro and in vivo assays to compare CAR constructs with distinct co-stimulatory domains and spacers.

The selection of STEAP1 as the target, owing to its expression in various malignancies, low/moderate expression in only a few normal tissue types, no expression in most normal tissues, and limited side effects of STEAP-1 targeting drugs used in clinical trials, add clinical relevance to the study.

The authors evaluated four CAR constructs, each featuring CD28 or 4-1BB co-stimulatory domains, coupled with either a CD8a or a mutated IgG spacer. The persistence and enhanced expansion observed in the CD8sp_41BBz variant suggest its potential superiority in achieving sustained anti-tumor efficacy. The findings of this study bear significant implications for the clinical translation of STEAP1-targeting CAR T cell therapies, however, the rapidly developing of CAR-T new generations reduced the novelty of this paper.

Questions are as follows:

Is there a special or better reason for the authors to employ retroviral delivery which is not commonly used in clinic?, perhaps a sentence to justify the use or comparison of vector system in discussion section will suffice.

Why did the authors choose CD8aTM to work with CD28 IC or 4-1BB IC rather than CD28TM.

Author Response

Please see the attached response letter
